# Pharmacoeconomics Aspects of Antiepileptic Drugs in Pediatric Patients with Epilepsy

**DOI:** 10.3390/ijerph19127517

**Published:** 2022-06-20

**Authors:** Dorota Kopciuch, Jędrzej Fliciński, Barbara Steinborn, Anna Winczewska-Wiktor, Anna Paczkowska, Tomasz Zaprutko, Piotr Ratajczak, Elżbieta Nowakowska, Krzysztof Kus

**Affiliations:** 1Department of Pharmacoeconomics and Social Pharmacy, Karol Marcinkowski University of Medical Sciences in Poznań, 60-865 Poznan, Poland; aniapaczkowska@ump.edu.pl (A.P.); tomekzaprutko@ump.edu.pl (T.Z.); p_ratajczak@ump.edu.pl (P.R.); kkus@ump.edu.pl (K.K.); 2Department of Developmental Neurology, Karol Marcinkowski University of Medical Sciences in Poznań, 60-355 Poznan, Poland; flicinski@hotmail.com (J.F.); bstein@ump.edu.pl (B.S.); awwiktor@ump.edu.pl (A.W.-W.); 3Department of Pharmacology and Toxicology Institute of Health Sciences, Collegium Medicum, University of Zielona Gora, 65-417 Zielona Góra, Poland; elapharm@ump.edu.pl

**Keywords:** antiepileptic drugs, children, epilepsy, pediatric patients with epilepsy, pharmacoeconomics, cost of pharmacotherapy, quality of life

## Abstract

Objective: This study assessed the differentiation of treatment costs with newer and older antiepileptic drugs (AEDs) through its correlation with treatment effectiveness and an adverse event (AE) in pediatric patients with epilepsy (PPE). Methods: PPE on monotherapy of AEDs for the last 6 months were screened for this study. Seizure frequency during the study was compared with that within 6 months before the study. The following parameters were also assessed: quality of life in epilepsy, Pittsburgh Sleep Quality Index, and Liverpool AEs Profile. An incremental cost-effectiveness ratio (ICER) analysis based on the costs of pharmacotherapy was also performed. Results: Out of 80 PPE, 67 completed the study, and 13 PPE were lost after failing to meet the inclusion criteria. A total of 56.71% of PPE were on newer AEDs, and 43.28% were on older AEDs. Newer and older AEDs did not differ significantly in seizure frequency reduction and quality of life parameters, although these were improved significantly during the study period. As per ICER, newer AEDs need an additional EUR 36.82 per unit reduction in seizure frequency. Conclusion: Newer AEDs have comparatively better efficacy, although not significantly better than older AEDs. However, the additional cost per unit improvement is quite high with newer AEDs, necessitating pharmacoeconomic consideration in pediatric epilepsy treatment.

## 1. Introduction

Epilepsy is the second most common neurological condition after headache. It is an important public health problem because of its burden on individuals and society in terms of morbidity and substantial economic burden on the health care system. It is characterized by recurrent seizures of cerebral origin [1].

The management of pediatric patients with epilepsy (PPE) comprises three main objectives: controlling seizures, avoiding treatment side effects, and maintaining or restoring quality of life (QOL) [2]. Most children with new-onset epilepsy achieve seizure freedom with appropriate antiepileptic drugs (AEDs). The fact that there is no clinical evidence to support a clear first-choice drug or an add-on drug for any given patient is one of the most significant challenges in treating children with epilepsy [3]. Moreover, predicting treatment response is not possible based on clinical features or laboratory results. Therefore, treatment selection should be individualized, and patients can be matched to a therapeutic regimen based on clinical profile, seizure type, and preference [4].

The pharmacological armamentarium for the management of epilepsy in children includes first-, second-, and third-generation AEDs [5]. Most of the second- and third-generation AEDs are licensed as an adjunctive treatment of epilepsy in adults and are, therefore, used off-label in pediatric populations on the basis of increasing evidence of their potential efficacy in children, especially in those older than 12 years of age.

Choosing the appropriate pharmacotherapy for children with epilepsy can have a huge impact, not only in terms of clinical outcomes but also on quality of life (QOL); children with epilepsy face many psychosocial challenges such as feelings of embarrassment, fearfulness, frustration, and helplessness that can negatively impact QOL [6,7,8]. Hence, appropriate AED use, along with monitoring of adverse effects and the assessment of QOL as an outcome measure, are important in the management of epilepsy to achieve optimal seizure control.

In addition to unmet treatment needs and the noted impact on patient QOL, epilepsy is associated with a substantial economic burden [9,10]. Most of the costs generated by children with epilepsy are connected with pharmacotherapy [11]. Approaches to pharmacotherapy management in epilepsy and pharmacotherapy cost analysis should also focus on cost-effectiveness outcomes, including the number of medications used, medication dose, and its impact on decreasing/increasing the number of seizure frequency and/or the medications’ generation and its effects on clinical parameters. Responses to this analysis could then be used to prioritize expenditures, make national as well as institutional decisions, and determine policy regarding epileptic patients [9].

The pharmacotherapy cost-effectiveness analysis is especially useful for the physicians who have to be bothered with the huge challenge of providing quality patient care with minimum cost for pediatric patients with epilepsy, as there is so much variation among different AEDs in terms of efficacy, quality of life, cost, and AE profiles. As limited studies are comparing the cost-effectiveness of different AEDs, pharmacoeconomic evaluation of the newer AEDs concerning older AEDs can help in understanding whether the incremental cost of newer AEDs is worth paying in terms of greater efficacy and safety. This study compared AED treatment efficacy, quality of life, and cost of treatment during a 6-month follow-up between newer and older AEDs administered as monotherapy to pediatric patients with epilepsy.

## 2. Methods

### 2.1. Study Group

The study was conducted at the Department of Developmental Neurology in Poland between 2019 and 2020. We evaluated patients that met the following inclusion criteria: aged ≤18 years, diagnosis of epilepsy for at least 1 year, a stable dose of AED for at least 6 months, and verbal consent to participation in the study.

After obtaining verbal informed consent, pediatric patients with epilepsy with the focal or generalized onset of seizures on monotherapy of newer or older AEDs over the last 6 months were screened for the study. Patients with other neurological, psychiatric, or chronic diseases (except epilepsy) and on that AED polytherapy were excluded from the study. Data regarding detailed demography, seizure history, prescribing information of AEDs, treatment efficacy parameters, and cost of treatment were collected using individual case report forms.

### 2.2. Treatment Efficacy Assessment

The efficacy of AED treatment was assessed prospectively through seizure control status and quality of life. The validated scales that were used in the study were the Quality of Life in Childhood Epilepsy Questionnaire (QOLCE-55) used to assess the quality of life (scale: 0–100; a higher score represents better quality of life), the Pittsburgh Sleep Quality Index (PSQI) used to the assessed quality of sleeping (scale: 0–21; a lower score represents better sleep quality), and the Liverpool Adverse Effect Profile (LAEP) used to assess AEs of AED treatment (scale: 0–76; a lower score represents fewer AEs).

Patients were interviewed twice during the study: at baseline and during a follow-up period. “Baseline” means measurements made at the beginning of the study. “Follow-up” means measurements made 6 months after the beginning of the study.

### 2.3. Calculation of Pharmacotherapy Costs

The data required for a pharmacotherapy regimen cost analysis were obtained retrospectively from the patients’ case histories, doctors’ request cards, and patients’ hospital discharge summaries. Mean pharmacotherapy costs were calculated based on a wholesale drug price list (average wholesale prices from three drug wholesalers were used), taking into account the exact doses of AEDs for every patient.

Pharmacotherapy costs for each patient were summed, and the mean was drawn. The average total cost of pharmacotherapy was based on a 6-month period. All amounts are specified in Euros as per the average exchange rates table of the National Bank of Poland as of 1 January 2019 (exchange rate: EUR 1.00 = PLN 4.3016) [12]. Total costs of pharmacotherapy refer to medicines taken 6 months after the beginning of the study.

A cost-effectiveness analysis of pharmacotherapy costs expressed as the incremental cost-effectiveness ratio (ICER) was conducted for the baseline and the follow-up period, based on the formula:ICER = (C_1_ − C_2_)/(E_1_ − E_2_)

C_1_ and E_1_ are the cost and effect of older AEDs, and C_2_ and E_2_ are the cost and effect of newer AEDs.

### 2.4. Statistical Analysis

Statistical analysis was performed using STATISTICA PL 13.0 (StatSoft). The data were expressed as the mean and SD. Significant differences between % of group results were determined by analysis of the test for proportions. The dependent *t*-test determined changes between baseline and follow-up for paired samples.

## 3. Results

### 3.1. Study Group

Out of 80 pediatric patients with epilepsy (PPE), 67 completed the study, and 13 PPE were lost after failing to meet the inclusion criteria. The 38 (56.71%) PPE were on newer AEDs, and 29 (43.28%) were on older AEDs (Table 1). Maximum numbers of PPE on newer AEDs were on levetiracetam (LEV) (34.32%), followed by lamotrigine (LTG) (20.68%) and vigabatrin (VGB) (17.68%). The most frequent older AEDs used by PPEs were valproate (VPA) (43.37%) and phenytoin (PHT) (26.31%) (Table 1).

The mean age of patients was 8.33 years, and the duration of the epilepsy was an average of 4.11 years (Table 1). In total, 43% of patients had generalized types of seizures and 16% had focal (Table 1).

### 3.2. Treatment Efficacy

It was observed that there was a major reduction in seizure frequency in all the PPE between the follow-up and baseline period (Table 2). Patients in the older AEDs group reported a decline from 9.93 to 7.76 in seizure frequency (*p* = 0.0106), whereas the newer AEDs group reported a decline from 7.98 to 4.72 (*p* < 0.0001) (Table 2).

A comparison between newer and older AEDs did not reveal any significant difference concerning QOLCE-55 when compared at baseline and follow-up measurements (Table 2). The mean QOLCE-55 score for PPE on older AEDs was 34.62 (±16.12) at baseline and 41.45 (±12.09) at follow-up (Table 2). A similar reduction in QOLCE-55 score was noted for newer AEDs, from 33.51 (±19.49) at baseline to 39.12 (±16.01) at follow-up (Table 2).

A significant difference between PSQI measurement in the older AEDs group (*p* = 0.0091) and the newer AEDs group (*p* = 0.0003) was observed. A decrease in PSQI score from 4.62 (±2.72) at baseline to 3.12 (±1.09) at follow-up among PPE on older AEDs was also observed. For PPE on newer AEDs, the score changed from 5.66 (±3.74) at baseline to 3.17 (±1.15) at follow-up (Table 2).

The mean LAEP score at baseline in PPE on older AEDs was 29.72 (±8.30), and it fell to 27.53 (±8.18) at follow-up (Table 2). A similar observation was made for patients with newer AEDs, in which the mean LAEP score decreased from 31.41 (±9.50) at baseline to 25.91 (±8.63) at follow-up. This decrease was statistically significant (*p* = 0.0101) (Table 2).

### 3.3. Incremental Cost-Effectiveness Ratio

The total average pharmacotherapy cost amounted to EUR 42.98 for PPE on older AEDs and EUR 83.12 for PPE on newer AEDs. No statistically significant difference between the number of average pharmacotherapy costs was observed. As per ICER analysis, EUR 36.82 will be the added cost for one extra unit reduction in seizure frequency for newer AEDs as compared with older AEDs (Table 3).

According to ICER, the added cost for a newer AEDs per unit improvement in the QOLCE-55 score would be EUR 32.90 (Table 3). Similarly, for PSQI and LAEP, an additional one unit improvement would have an added cost of EUR 40.54 and 12.12 for newer AEDs (Table 3).

## 4. Discussion

Out of 67 pediatric patients with epilepsy enrolled in our study, 56.71% were prescribed newer AEDs, i.e., LEV and OXC, in focal and generalized epilepsy. This is in accordance with a previous study conducted in India [13]. LEV constituted a major part of all AEDs considered in the study, as it was prescribed to 32.32% of patients. The studies and trials conducted previously depicted a high proportion of LEV prescriptions that led to consideration by the National Institute for Health and Care Excellence (NICE) to include LEV as a potential first-line agent in focal seizures and adjunctive in generalized seizures [14,15].

According to studies, children diagnosed with epilepsy experience changes in sleep quality, sleep architecture, sleep latency, and spontaneous awakenings. They also suffer from sleep fragmentation and daytime sleepiness more often [16,17,18]. Unfortunately, sleep complaints are rarely voiced during a pediatric visit and are often misdiagnosed in children with epilepsy. It has been suggested that sleep abnormalities and changes in children can affect seizure control, behavior, neuropsychological development, school performance, and even family relationships. This means diagnosis and treatment of sleep problems may contribute to better clinical seizure control [18]. According to one study, some AEDs can improve sleep stability [19]. This poses the question of whether the improvement in sleep patterns is a direct consequence of AED use or a consequence of epileptic symptoms being suppressed. Furthermore, epilepsy is recognized to affect physical, mental, and social functioning. This issue mainly affects children with epilepsy, who, according to research, share a number of problems not only related to epileptic seizures. These include cognitive problems; they also experience difficulties at school and are exposed to social stigma among peers [19,20,21]. An average score of quality of life at 46.82 ± 10.90 was obtained in a QOLCE-55 questionnaire-based study conducted among children in India. A significantly higher degree of cognitive impairment was observed in children with epilepsy. Patients on older AEDs were significantly more satisfied with their quality of life.

Our results of QOLCE-55 and other quality of life (PSQI and LAEP) improvement with time, i.e., after 6 months of AEDs treatment, are supported by the findings of previous studies [13,15]. However, as in our study, these studies could not find any significant difference between newer and older AEDs. It was observed that older AEDs decreased the seizure frequency up to 2.17, whereas newer AEDs decreased by 3.26 after 6 months of AEDs treatment. Earlier reports in the literature raised doubts as to the superiority of newer AEDs in terms of efficacy and their long-term benefits. They failed to detect any significant differences in efficacy outcomes [22,23]. No significant difference was found between both groups in PSQI and LAEP scores [24,25]. The previous study conducted on older adults reported that there was no difference in LEAP scores between newer and older AEDs [26].

The total average costs of pharmacotherapy were assessed at EUR 42.98 per patient treated with older AEDs and EUR 83.12 per patient on newer AEDs. This was similar to a study conducted in India, where the total annual median cost of medicines was USD 82.24 for the newer AEDs group. This was quite a lot higher than the cost of older AEDs (USD 14.76). In a study conducted in Spain, the cost of second-generation AEDs was 61.8% of the total costs of AEDs, and third-generation AEDs represented 27.3% of the total costs of pharmacotherapy [27]. Other past studies on drug acquisition costs carried out in developed countries showed a wide variation in the contribution of direct medical cost to the total cost of treatment [28,29]. The cost of medicine constituted 96.19% of direct medical costs and 73.33% of the total cost of treatment. In the past, various studies have highlighted the contribution of the cost of drugs to the total direct cost.

This cost-effectiveness analysis found that newer AEDs are more expensive and perform comparably better, requiring further analysis to determine the potential cost-effectiveness among newer and older AEDs. Cost-effectiveness analysis is a complete pharmacoeconomic analysis that identifies, measures, evaluates, and compares the costs and outcomes of alternative health programs. In the cost-effectiveness analysis, the results are presented in natural units such as cure rate, life years gained, and time free from disease symptoms. The condition for conducting this type of analysis is the use of the same unit of measurement of the result for the compared health programs. It is one of the broadest types of analysis, so we decided to perform it.

We also assumed that it would be very informative to conduct a cost–utility analysis, but we had limited data to provide it. To perform a cost–utility analysis, apart from the cost of treating newer and older AEDs, QALY calculations are also needed. In turn, to calculate QALY, apart from the quality of life, we also need to know the number of years of life after using newer and older AEDs. Unfortunately, we do not have such data. Hence, we limited ourselves to the presented results [30]. This was also established from an ICER analysis that showed that newer AE drugs have higher additional costs to achieve one unit of additional health score in terms of reduction in seizure frequency, better quality of life, sleep quality, and reduction in AE compared with older AEDs. However, whether the added cost is high or low depends upon the ICER ceiling value of a given country. ICER data from the current study can help in making decisions in the Polish health care system as individualized or individual, based on socioeconomic conditions. It is also worth adding that clinical and economic analyses are an excellent source of reliable information for clinicians, providing knowledge on how to make rational decisions. However, to make it possible, clinicians should be trained in the proper use of this type of analysis. A good place for this is to expand the meaning of the teaching hospitals, in which activities include not only the provision of health care but also teaching and under- and postgraduate training [31,32].

To the best of our knowledge, this study is the first study in Poland to assess, compare, and correlate improvement in the quality of life and cost of AED pharmacotherapy for pediatric patients with epilepsy. However, this study has certain limitations such as a small sample size and short duration to measure the change in the intangible health-related measures. Other intangible components should be considered for further analysis.

## 5. Conclusions

Based on this study, it can be concluded that the treatment efficacy parameters based on the quality of life and seizure reduction show comparatively greater improvements with newer AEDs compared with older AEDs. A cost analysis has also shown that treating newer AEDs is more costly than older AEDs. It should be noted, however, that the study was conducted on a relatively small study group, and these results may be much more favorable in a larger cross-sectional group. However, the clinical and economic study of the cost analysis of pharmacotherapy, taking into account AED generation and the regimen, should become the subject of more detailed considerations, which will ultimately provide health care decision-makers and clinicians with reliable tools to make rational decisions. In addition, the above research results, together with the socioeconomic conditions, may help in the individualization of epilepsy treatment in the Polish population.

## Figures and Tables

**Table 1 ijerph-19-07517-t001:** Demographic and clinical data (*n* = 67).

Age; average ± SD	8.33 ± 4.37
Duration of epilepsy, years (average ± SD)	4.11 ± 1.22
Type of seizures; *n* (%)	
generalized	43 (64.17)
focal	24 (35.83)
Type of AEDs; *n* (%)	
OLDER:	29 (43.28)
Valproate	(43.37%)
Phenytoin	(26.31%)
Carbamazepine	(19.80%)
Clobazam	(10.52%)
NEWER:	38 (56.71)
Levetiracetam	(32.32%)
Vigabatrin	(17.68%)
Oxcarbazepine	(14.27%)
Topiramate	(12.42%)
Lacosamide	(2.63%)
Lamotrigine	(20.68%)

SD, standard deviation; AEDs, antiepileptic drugs.

**Table 2 ijerph-19-07517-t002:** Changes in the treatment efficacy parameters considered in the study (*n* = 67).

	Older AEDs (*n* = 29)			Newer AEDs (*n* = 38)			*p*-Value Older AEDs vs. Newer AEDs at Baseline/Follow-Up	
	Baseline	Follow-Up	*p*-Value	Baseline	Follow-Up	*p*-Value	Baseline	Follow-Up
**Average seizure frequency per month** **(mean ± SD)**	9.93 (3.26)	7.76 (2.98)	0.0106	7.98 (3.88)	4.72 (1.23)	<0.0001	0.0328	<0.00001
**QOLCE-55 (mean ± SD)**	33.51 (19.49)	39.12 (16.01)	0.1747	34.62 (16.12)	41.45 (12.09)	0.0738	0.8045	0.5155
**PSQI (mean ± SD)**	4.62 (2.72)	3.12 (1.09)	0.0091	5.66 (3.74)	3.17 (1.15)	0.0003	0.2110	0.8575
**LAEP (mean ± SD)**	29.72 (8.30)	27.53 (8.18)	0.3160	31.41 (9.50)	25.91 (8.63)	0.0101	0.4492	0.4391

SD, standard deviation; AEDs, antiepileptic drugs; QOLCE-55, Quality of Life in Childhood Epilepsy Questionnaire; PSQI, Pittsburgh Sleep Quality Index; LAEP, Liverpool Adverse Effect Profile.

**Table 3 ijerph-19-07517-t003:** Incremental cost-effectiveness ratio value between the patients on older and newer AEDs.

	Older AEDs	Newer AEDs	Difference	ICER
**The total average cost of pharmacotherapy in EUR (±SD)**	42.98 (±9.12)	83.12 (±12.03)	40.14 (±8.90)	
**Change of average seizure frequency per month for 6 months**	2.17	3.26	1.09	36.82 ^#^
**Improvement in QOLCE-55 mean scores at the end of the study as compared with enrollment**	5.61	6.83	1.22	32.90 ^#^
**Improvement in PSQI mean scores at the end of the study as compared with enrollment**	1.5	2.49	0.99	40.54 ^#^
**Improvement in LAEP mean scores at the end of the study as compared with enrollment**	2.19	5.5	3.31	12.12 ^#^

^#^ Added cost for newer AEDs to improve the parameter by one unit as compared with conventional AEDs; AEDs, antiepileptic drugs; QOLCE-55, Quality of Life in Childhood Epilepsy Questionnaire; PSQI, Pittsburgh Sleep Quality Index; LAEP, Liverpool Adverse Effect Profile; ICER, incremental cost-effectiveness ratio.

## Data Availability

Not applicable.

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
