# Peer review of "Pharmacoeconomics Aspects of Antiepileptic Drugs in Pediatric Patients with Epilepsy"

_ijerph, 2022, doi:10.3390/ijerph19127517_

Round 1

Reviewer 1 Report

Referee Report

IJERPH Manuscript #1699890
Pharmacoeconomics aspects of antiepileptic drugs in pediatric patients with epilepsy

Summary

This paper looks at the cost effectiveness of newer vs older anti-epileptic drugs (AEDs) for pediatric patients with epilepsy, using data for a small number of patients in Poland. Overall, they seem to find that costs for newer drugs are not that much higher, but that effectiveness is higher (with the exception of QOL, for which older AEDs seem to work better). While I think this paper could be useful to clinicians, and of interest to some readers, I have some major concerns about some of the methods and presentation of results. Some major editing for English language would be helpful, too. If these concerns can be addressed, the paper could be a valuable contribution to the literature.

Comments:

(1)   Need to spell out acronyms in abstract – i.e., AED and AE

(2)   Introduction

a.      Can’t have a paragraph that is only 1 sentence long – add some more information to the first sentence of the introduction, particularly more info about what makes epilepsy an important topic to study especially wrt cost analyses

b.      Combine 2nd & 3rd paragraphs (lines 33-42)

c.      Combine 4th & 5th paragraphs (lines 43-48)

d.      Edit sentence lines 49-54 to read “Choosing the appropriate pharmacotherapy for children with epilepsy can have huge impact, not only in terms of clinical outcomes, but also on quality-of-life (QOL); children with epilepsy face many psychosocial challenges such as <<give examples of psychosocial challenges here>> that can negatively impact QOL. Hence, when administering AEDs, it is important to monitor not just clinical outcomes like seizure control and adverse effects, but also QOL.”

e.      Sentence starting on lines 54-57 about economic burden of epilepsy should be a new paragraph – may be able to combine with final paragraph of the intro

f.       Need to provide a little more info about the literature on CEAs of AEDs either overall and/or in context of PPEs

(3)   Methods

a.      Line 69- replace “one of the departments” with “a department”

b.      Line 70 – take out word ‘consectutive’

c.      Section 2.3 – calculation of pharmacotherapy costs:

                                                    i.     A lot more information is needed in this subsection. The authors describe where they get variables from, but not how they are calculating costs exactly. Are they just multiplying the # of prescriptions/doses by the WSP? So no other appointments/costs are being incorporated?

                                                   ii.     Related to this – authors write (lines 92-93) “..data obtained ..from .. patient’s case histories, doctors’ request cases, and patient hospital discharge summaries”. I am not familiar with these data sources – I’m not sure if it is a language issue. Are these electronic health record data? Regardless, more detail on the exact variables taken from these sources need to be given. Also, more info needed on why these data sources were used and/or how AEDs are prescribed. For example, will all AED prescriptions be contained in these files, or could the authors/data be missing some? Could it be possible that the authors are seeing AEDs prescribed, but not prescriptions fulfilled.

                                                  iii.     Cost – again, I am not familiar with the wholesale drug price list as a data source – where exactly are they getting this data from and do the authors mean Average Wholesale Price? If so, this is likely not the actual price paid by patients, so some discussion of this needs to be made, likely both here in the methods section as well as in the results section.

                                                  iv.     Outcome variables –you need to explain what you’re doing in Table 2 (see comments below). Where are the p-values coming from in columns 3 and 6 – what kind of test? Also, I don’t understand what you’re doing in columns 7 & 8 – are those the p-values for the differences in baseline values (col 7) and f/u values (col 8)? Why do you provide those?

(4)   Results

a.      Revise beginning of sentence (lines 105-106) to be “The most frequent newer AEDs used by PPEs were …… “

b.      Line 138-9 “Average drug cost was EUR 42.98 for PPE on older AEDs …” – is that the total cost for the whole 6 month supply of AEDs? That seems really low – is that typical?

(5)   Tables

a.      Table 1 – justify category headings (Types of Seizures, Types of AEDs, etc) to the left, and values (generalized, focus; valproate, etc) to the right

b.      Table 2 – still need to report the p-value, even if not significant.

c.      Table 3 – Do you have the sign for row 3 (improvement in QOLCE 55) incorrect? – if older AEDs seem to have a better impact on QOL, shouldn’t the # should actually be -1.22 and the ICER should have a negative sign, too?

(6)   Discussion

a.      2nd paragraph (lines 160-180) contains a lot of clinical information not necessarily relevant to this paper – foe example, lines 170-173 “This suggests  a need … “ – = what does the relating of clinical values to polysomnographic analysis of sleep structure have to do with the cost effectiveness of AEDs? Consider dropping that sentence and otherwise re-wiorking this paragraph.

b.      4th paragraph,

                                                    i.     line 196- “This was quite a lot higher than the cost of older AEDs.” This relates to my comment about costs above (4.b): if the cost difference (83.12 vs 42.98) between newer and older drugs is the average total cost for the whole 6 months – which is how the cost-effectiveness analysis is indicating it is – then this seems like a small amount, not a lot. ~EUR 40 over 6 months is a lot?

                                                   ii.     The rest of this paragraph is also confusing (ie, mixes discussion of cost of newer AEDs as % of total AED costs with discussion of cost of AEDs as % of total treatment costs) and needs to be reworked. 

c.      5th paragraph

                                                    i.     why do the authors conclude that more analysis is needed to determine the cost-effectiveness? They should have all the info they need here to make that determination from their data. See comments under #7 below

                                                   ii.     The authors repeat the mistake here (see comments about Table 3 above) that they find newer AEDs to be more expensive and more effective – based on my reading of QOL, the older AEDs are more effective in that dimension          

d.      More information in the discussion (or maybe just collating the info into one paragraph as it may just be spread out over the section) is needed putting the authors results in the context of the literature – have there been any studies from any countries looking at the CEA of AEDs for PPEs? What about for any patient group? Where do your results fall in that literature?

(7)   Overall

a.      Since you have QOL scores, why are you not doing a cost-utility analysis looking at the Cost/QALY?

b.      Why are you not generating a WTP curve for AEDs, to see where your estimated ICER falls – you should be able to do this through modeling.

Author Response

Summary

This paper looks at the cost effectiveness of newer vs older anti-epileptic drugs (AEDs) for pediatric patients with epilepsy, using data for a small number of patients in Poland. Overall, they seem to find that costs for newer drugs are not that much higher, but that effectiveness is higher (with the exception of QOL, for which older AEDs seem to work better). While I think this paper could be useful to clinicians, and of interest to some readers, I have some major concerns about some of the methods and presentation of results. Some major editing for English language would be helpful, too. If these concerns can be addressed, the paper could be a valuable contribution to the literature.

 Comments:

(1)   Need to spell out acronyms in abstract – i.e., AED and AE

Response: I'm sorry for overlooking it. Thank you for your comment. The information was added as you suggest.

(2)   Introduction

  1. Can’t have a paragraph that is only 1 sentence long – add some more information to the first sentence of the introduction, particularly more info about what makes epilepsy an important topic to study especially wrt cost analyses

Response: Thank you for the comment. Some information was added.

  1. Combine 2nd & 3rd paragraphs (lines 33-42)

Response: Corrected, as you suggest.

  1. Combine 4th & 5th paragraphs (lines 43-48)

Response: Corrected, as you suggest.

  1. d. Edit sentence lines 49-54 to read “Choosing the appropriate pharmacotherapy for children with epilepsy can have huge impact, not only in terms of clinical outcomes, but also on quality-of-life (QOL); children with epilepsy face many psychosocial challenges such as <<give examples of psychosocial challenges here>> that can negatively impact QOL. Hence, when administering AEDs, it is important to monitor not just clinical outcomes like seizure control and adverse effects, but also QOL.”

Response: Thank you for your advice. Corrected as you suggest.

  1. Sentence starting on lines 54-57 about economic burden of epilepsy should be a new paragraph – may be able to combine with final paragraph of the intro

Corrected, as you suggested.

  1. f. Need to provide a little more info about the literature on CEAs of AEDs either overall and/or in context of PPEs

Response: The information was added, as you suggested.

(3)   Methods

  1. Line 69- replace “one of the departments” with “a department”

Response: corrected

  1. Line 70 – take out word ‘consectutive’

Response: corrected

  1. Section 2.3 – calculation of pharmacotherapy costs:

  1. A lot more information is needed in this subsection. The authors describe where they get variables from, but not how they are calculating costs exactly. Are they just multiplying the # of prescriptions/doses by the WSP? So no other appointments/costs are being incorporated?

Response: Pharmacotherapy costs for each patient were summed and the mean was drawn. The average total cost of pharmacotherapy is based on a 6-month period.

This information has been added to the manuscript as requested by you.

A number of other direct medical and non-medical costs were also analyzed in the study group, but these have already been published previously. The presented work concerns only the costs of pharmacotherapy, which deserves particular attention in patients with epilepsy.

  1. Related to this – authors write (lines 92-93) “..data obtained ..from .. patient’s case histories, doctors’ request cases, and patient hospital discharge summaries”. I am not familiar with these data sources – I’m not sure if it is a language issue. Are these electronic health record data?

Response: Thanks for the question.

These are data in both electronic and paper forms. In Poland, although the completion of electronic documentation is obligatory, doctors usually provide more detailed information in the paper documentation of patients, therefore it was decided to analyze both types of documentation for each patient.

Regardless, more detail on the exact variables taken from these sources need to be given. Also, more info needed on why these data sources were used and/or how AEDs are prescribed. For example, will all AED prescriptions be contained in these files, or could the authors/data be missing some? Could it be possible that the authors are seeing AEDs prescribed, but not prescriptions fulfilled.

      iii.     Cost – again, I am not familiar with the wholesale drug price list as a data source – where exactly are they getting this data from and do the authors mean Average Wholesale Price? If so, this is likely not the actual price paid by patients, so some discussion of this needs to be made, likely both here in the methods section as well as in the results section.

Response: Thank you for your comment. Agree, the information should be clarified.

That's right- Average Wholesale Prices from three drug wholesalers were used. The study adopts the social perspective, i.e. the broadest possible, which also takes into account the costs incurred by patients; it is a costing methodology in line with the guidelines of pharmacoeconomic research.

Nevertheless, you are right - the information should be clarified. The data has been added.

  1. iv. Outcome variables –you need to explain what you’re doing in Table 2 (see comments below). Where are the p-values coming from in columns 3 and 6 – what kind of test?

The data were added, as you suggested

Also, I don’t understand what you’re doing in columns 7 & 8 – are those the p-values for the differences in baseline values (col 7) and f/u values (col 8)? Why do you provide those?

Response: That's right- column 7 shows P-value scores between baseline values (older AEDs at baseline vs. newer AEDs at baseline) and column 8 P values for follow-up (older AEDs at follow-up vs. newer AEDs at follow-up). This information was to show whether there are differences in the analyzed parameters at the beginning of the study and whether they changed after the follow-up period.

(4)   Results

  1. Revise beginning of sentence (lines 105-106) to be “The most frequent newer AEDs used by PPEs were …… “

Response: corrected

  1. Line 138-9 “Average drug cost was EUR 42.98 for PPE on older AEDs …” – is that the total cost for the whole 6 month supply of AEDs? That seems really low – is that typical?

Response: This is the average cost per patient/ 6 months. There were patients whose cost was, for example, EUR 150, but there were also patients whose cost was EUR 30. It depends on the number of drugs used, their dose, frequency of administration, and, ultimately, the price of the drug. Please also note that the price of the older generation drug is usually lower than newer. Nevertheless, thank you for the apt remark and your attention.

(5)   Tables

  1. a. Table 1 – justify category headings (Types of Seizures, Types of AEDs, etc) to the left, and values (generalized, focus; valproate, etc) to the right

Response: I’m not sure if I understand your questions. All subheadings in the table are centered.

 I think the esthetic features of the tables are proper.

  1. Table 2 – still need to report the p-value, even if not significant.

Response: The data were added, as you sugesst.

  1. c. Table 3 – Do you have the sign for row 3 (improvement in QOLCE 55) incorrect? – if older AEDs seem to have a better impact on QOL, shouldn’t the # should actually be -1.22 and the ICER should have a negative sign, too?

Response: Sorry for the mistake and thank you for your attention. By mistake, I included incorrect data in the wrong column. Newer AEDs produced a better quality of life satisfaction, admittedly slightly higher, but still. The data were corrected.

(6)   Discussion

  1. a. 2nd paragraph (lines 160-180) contains a lot of clinical information not necessarily relevant to this paper – foe example, lines 170-173 “This suggests a need … “ – = what does the relating of clinical values to polysomnographic analysis of sleep structure have to do with the cost effectiveness of AEDs? Consider dropping that sentence and otherwise re-wiorking this paragraph.

Response: The sentence has been deleted.

  1. 4th paragraph,

  1. line 196- “This was quite a lot higher than the cost of older AEDs.” This relates to my comment about costs above (4.b): if the cost difference (83.12 vs 42.98) between newer and older drugs is the average total cost for the whole 6 months – which is how the cost-effectiveness analysis is indicating it is – then this seems like a small amount, not a lot. ~EUR 40 over 6 months is a lot?

Full-sentence quoted "This was similar to a study conducted in India where the total annual median cost of medicines was USD 82.24 for the newer AEDs group. This was quite a lot higher than the cost of older AEDs (USD 14.76)." for a study conducted in India.

  1. The rest of this paragraph is also confusing (ie, mixes discussion of cost of newer AEDs as % of total AED costs with discussion of cost of AEDs as % of total treatment costs) and needs to be reworked.

I made some changes. Nevertheless, in the opinion of the authors, the current content of the paragraph accurately describes the place in the total cost structure of pharmacotherapy costs in relevant reports.

  1. 5th paragraph

  1. why do the authors conclude that more analysis is needed to determine the cost-effectiveness? They should have all the info they need here to make that determination from their data. See comments under #7 below

Response: Yes, our study has proven that newer AEDs are showing good results, but it should be noted, that the study was conducted by a relatively small study group, and these results may be much more favorable in a larger cross-sectional group. However, the clinical and economic study of the cost analysis of pharmacotherapy, taking into account AEDs' generation and the regimen should become the subject of more detailed considerations, which will ultimately provide health care decision-makers and clinicians with reliable tools to make rational decisions.

This information was added in the conclusion.

  1. The authors repeat the mistake here (see comments about Table 3 above) that they find newer AEDs to be more expensive and more effective – based on my reading of QOL, the older AEDs are more effective in that dimension  

 Response:  Sorry for the mistake and thank you for your attention. By mistake, I included incorrect data in the wrong column. Newer AEDs produced a better quality of life satisfaction, admittedly slightly higher, but still. The data were corrected.

  1. More information in the discussion (or maybe just collating the info into one paragraph as it may just be spread out over the section) is needed putting the authors results in the context of the literature – have there been any studies from any countries looking at the CEA of AEDs for PPEs? What about for any patient group? Where do your results fall in that literature?

Response: To the best of our knowledge, this study is the first study in Poland to assess, compare and correlate improvement in the quality of life and cost of AEDs’ pharmacotherapy for pediatric patients with epilepsy.

(7)   Overall

  1. Since you have QOL scores, why are you not doing a cost-utility analysis looking at the Cost/QALY?
  2. Why are you not generating a WTP curve for AEDs, to see where your estimated ICER falls – you should be able to do this through modeling.

Thank you for your suggestion. We agree that it would be interesting to analyze it and we will certainly consider it in further research.

Reviewer 2 Report

Pharmacoeconomic evaluation of antiepileptic drugs is a very important issue and helps in understanding whether the incremental cost of newer AEDs is worth paying in terms of greater efficacy and safety.

The manuscript is rather interesting and the research has been thoroughly prepared and conducted. 

However, it is worth adding a few sentences about the statistics in Tab.2, i.e. what was the type of tests (it looks like the dependent t-test for paired samples), and which statistical program was used to obtain the results (p-value). How to interpret NS, what was the value of the p-value?

In Tab.3 'The total average cost of pharmacotherapy in EUR (±SD)' - I can not see SD and I want to know how 'the total average cost' was measured (per month/per year/per 6-month treatment and moreover there were different drugs and different percentage/frequency of drug use).

How looks the formula for calculating ICER in Tab.3?

The third row ('Improvement in QOLCE-55') is questionable - the difference for older AEDs is greater than for newer AEDs. It should be explained in text because if we only look at this parameter it means that we pay more for worse (subjectively) results.

In summary, when analyzing costs, it is also worth considering the comparison of potential side effects of the drugs used.

Author Response

Pharmacoeconomic evaluation of antiepileptic drugs is a very important issue and helps in understanding whether the incremental cost of newer AEDs is worth paying in terms of greater efficacy and safety.

The manuscript is rather interesting and the research has been thoroughly prepared and conducted.

  1. However, it is worth adding a few sentences about the statistics in Tab.2, i.e. what was the type of tests (it looks like the dependent t-test for paired samples), and which statistical program was used to obtain the results (p-value). How to interpret NS, what was the value of the p-value?

Response: The data were added.

  1. In Tab.3 'The total average cost of pharmacotherapy in EUR (±SD)' - I can not see SD

Response:  I'm sorry for my mistake. Data has been added, as you suggest.

  1. I want to know how 'the total average cost' was measured (per month/per year/per 6-month treatment

Response: Thank you for your comment. As in the chapter "methods" (page 3, lines 97-98) - "The total cost of pharmacotherapy applies to drugs taken 6 months after the start of the study", means it was calculated by summing the patients' pharmacotherapy and averaging per patient, while "the average total cost of pharmacotherapy ”refers to the entire measure period, up to 6 months.

  1. moreover there were different drugs and different percentage/frequency of drug use).

Response: That's right. This has been counted patient by patient, tablet by tablet.

  1. How looks the formula for calculating ICER in Tab.3?

Response:

ICER= (C1-C2)/(E1-E2)

C1 and E1 are the cost and effect of older AEDs, and  C2 and E2 are the cost and effect of newer AEDs.

The information was added in the „methods” section.

  1. The third row ('Improvement in QOLCE-55') is questionable - the difference for older AEDs is greater than for newer AEDs. It should be explained in text because if we only look at this parameter it means that we pay more for worse (subjectively) results.

Response: Sorry for the mistake and thank you for your attention. By mistake, I included incorrect data in the wrong column. Newer AEDs produced better quality of life satisfaction, admittedly slightly higher, but still.

  1. In summary, when analyzing costs, it is also worth considering the comparison of potential side effects of the drugs used.

Response: Analysis of the ADRs was the subject of another my publication, so I could not include it again in the manuscript presented to you. Nevertheless, I agree with you - it would be valuable to the reader to combine this information in one manuscript.
